

# Validation of reference genes for gene expression studies in post-harvest leaves of tea plant (*Camellia sinensis*)

Zi-wei Zhou[1,2], Hui-li Deng[1,2], Qing-yang Wu[1,2], Bin-bin Liu[1,2], Chuan Yue[1], Ting-ting Deng[1], Zhong-xiong Lai[2] and Yun Sun[1]

[1] Key Laboratory of Tea Science in Fujian Province, College of Horticulture Fujian Agriculture and Forestry University, Fuzhou, P. R. China
[2] Institute of Horticultural Biotechnology, Fujian Agriculture and Forestry University, Fuzhou, P. R. China

## ABSTRACT

Tea is one of three major non-alcoholic beverages that are popular all around the world. The economic value of tea product largely depends on the post-harvest physiology of tea leaves. The utilization of quantitative reverse transcription polymerase chain reaction is a widely accepted and precise approach to determine the target gene expression of tea plants, and the reliability of results hinges on the selection of suitable reference genes. A few reliable reference genes have been documented using various treatments and different tissues of tea plants, but none has been done on post-harvest leaves during the tea manufacturing process. The present study selected and analyzed 15 candidate reference genes: *Cs18SrRNA*, *CsGADPH*, *CsACT*, *CsEF-1α*, *CsUbi*, *CsTUA*, *Cs26SrRNA*, *CsRuBP*, *CsCYP*, *CselF-4α*, *CsMON1*, *CsPCS1*, *CsSAND*, *CsPPA2*, *CsTBP*. This study made an assessment on the expression stability under two kinds of post-harvest treatment, turn over and withering, using three algorithms—GeNorm, Normfinder, and Bestkeeper. The results indicated that the three commonly used reference genes, *CsTUA*, *Cs18SrRNA*, *CsRuBP*, together with *Cs26SrRNA*, were the most unstable genes in both the turn over and withering treatments. *CsACT*, *CsEF-1α*, *CsPPA2*, and *CsTBP* were the top four reference genes in the turn over treatment, while *CsTBP*, *CsPCS1*, *CsPPA2*, *CselF-4α*, and *CsACT* were the five best reference genes in the withering group. The expression level of lipoxygenase genes, which were involved in a number of diverse aspects of plant physiology, including wounding, was evaluated to validate the findings. To conclude, we found a basis for the selection of reference genes for accurate transcription normalization in post-harvest leaves of tea plants.

## INTRODUCTION

The quantitative real-time polymerase chain reaction (RT-qPCR) is being used widely as a preferred and powerful approach applied to detect gene expression levels in molecular biology based on the polymerase chain reaction (PCR) (*Peters et al., 2004*; *Zhang et al., 2009*). According to the different methods of calculation, RT-qPCR can be divided into two

Corresponding author
Yun Sun, sunyun1125@126.com

categories: absolute and relative quantification (*Lee et al., 2008*). In contrast to absolute quantification, relative quantification utilizes a relatively stable control gene as a reference. Although many reference genes are expressed at relatively constant levels under most situations of biotic and abiotic stress, such as *LDHA*, *NONO*, and *PPIH*, they could change based on different experimental conditions (*Keshishian et al., 2015*). An important impact part of the RT-qPCR assay is the selection of a reliable reference gene to normalize the result as this determines the accuracy of the assay results.

Post-harvest handling is a vital process to promote the business value of horticultural crops (*Brash, 2007*; *Cantwell & Kasmire, 1992*). Plenty of research has shown that when a plant organ is harvested, its life processes continue to produce changes and before it becomes unmarketable, several biochemical processes continuously change its original composition through numerous enzymatic reactions (*Sun et al., 2012*). Simultaneously, regulatory genes still play important roles during post-harvest handling (*Blauer et al., 2013*), and the selection of suitable reference genes to normalize the expression levels of target genes has become significant. Unlike horticultural crops, the post-harvest handling of tea leaves, such as instant water-loss under sunlight (*Zhang et al., 2012*) or mechanical force by equipment (*Guo et al., 2016*), can accelerate chemical change and even create more physical damage. Thus far, appropriate reference gene selections have been reported for horticultural crops, such as roses (*Meng et al., 2013*), apples (*Storch et al., 2015*), bananas (*Chen et al., 2011*), longans (*Wu et al., 2016b*), papayas (*Zhu et al., 2012*), and grapes (*González-Agüero et al., 2013*), under different post-harvest conditions. Gene expressions of tea leaves have been detected under conditions such as withering under different light qualities (*Fu et al., 2015*; *Xiang et al., 2015*), wounding by tossing (*Gui et al., 2015*), and the manufacturing process (*Cho et al., 2007*). However, the validation of reference genes of tea plants has concentrated almost entirely on organs and tissues (*Sun et al., 2010*), species (*Gohain et al., 2011*), metallic stress (*Wang et al., 2017*), and hormonal stimuli (*Wu et al., 2016a*), there is no report about the selection of reference genes during post-harvest conditions as yet.

The tea plant is an important cash crop in many countries. The post-harvest leaves of tea plants determine the business value of final tea products (*Pothinuch & Tongchitpakdee, 2011*). Tea leaves, which are rich in polyphenols, amino acids, alkaloid vitamins, and minerals, have a profound health and nutrition value (*Stagg & Millin, 2010*; *Sun, Lin & Lü, 2004*). Although plucked from the tea tree, tea leaves still maintain certain enzymes, such as cellulase (*Wang, Yan & Yao, 2012*), which remains active for a period of time. Isolated signals will induce some enzymes and relative genes to change under oxidizing, wounding, and water-loss conditions, and then secondary metabolites render the differences (*Ramdani, Chaudhry & Seal, 2013*), especially oolong tea which was considered to have most complex process (*Sakata et al., 2005*).

In order to explore the stability of the reference genes of tea plants reported in previous studies in post-harvest treatment, we planned to screen the most suitable single reference and multiple reference genes in post-harvest treatment. We carried out the following studies. In the current study, one bud and three leaves of the tea plant (*Camellia sinensis cv.* Huangdan) were placed under a series of external mechanical forces and water-loss

as test materials. A total of 15 reference genes of the tea tree, including 18S ribosomal RNA (18SrRNA), glyceraldehyde-3-phosphate (GADPH), actin (ACT), elongation factor-1α (EF-1α), ubiquitin protein (Ubi), tubulin alpha (TUA), 26S ribosomal RNA (26SrRNA), rubisco bis phosphatase (RuBP), cyclophilin (CYP), eukaryotic translation (elF-4α), monensin sensitivity1 (MON1), phytochelatin synthase (PCS1), family protein gene (SAND), protein phosphatase 2A gene (PPA2), and the TATA-box binding protein gene (TBP) were selected due to previous evidence of their stable expression (*Sun et al., 2010*; *Gohain et al., 2011*; *Wang et al., 2017*; *Wu et al., 2016a*; *Pothinuch & Tongchitpakdee, 2011*). Based on previous studies, three publicly available software tools, GeNorm v3.5 (*Vandesompele et al., 2002*), NormFinder v0.953 (*Heimpel et al., 2010*) and Bestkeeper v1.0 (*Barsalobrescavallari et al., 2009*), were selected to rank the stability of the 15 candidate reference genes. To verify the study results, the expression level of the lipoxygenase (LOX) gene (*CsLOX1*) was detected under post-harvest treatment normalized to the most stable and unstable genes, as *CsLOX1* could respond to mechanical wounding and act a key role in some characteristic volatile compounds during tea processing (*Gui et al., 2015*; *Zhou et al., 2017*). The results might provide an important reference for work involving the selection of suitable reference genes under different experimental conditions in the post-harvest leaves of the tea plant.

## MATERIALS AND METHODS

### Plant material and post-harvest treatment

Tea leaves were collected from *C. sinensis cv.* Huangdan, a main and popular cultivar in oolong tea production area, in educational practicing base (26°04′N, 119°14′E) of Fujian Agriculture and Forestry University (Fuzhou, China) from 4–5 p.m. on July 27, 2017 under sunny conditions. "One bud and three leaves" means one bud and three leaves on the same branch (Fig. 1), which is commonly used as a whole for oolong tea manufacturing in China. The post-harvest treatment involved the usual methods and stimulation that are used in the oolong tea manufacturing process, which was typical for process industrially (*Gui et al., 2015*). The fresh leaves (F) were withered under gentle sunlight (25 °C, 120,000 Lux) for 30 min. Subsequently, half of withered leaves (500 g) were shaken three times for 5 min, hourly (T1–T3) until de-enzyming. As is presented from Fig. 1B, T1, T2, T3 were sampled after first time, second time, and third time turn over, respectively. The remaining withered leaves was used to set a control group without turn over, we sampled CK1–CK3 at the same time point. All treatments were performed at 24 °C, with a relative humidity of 45%, and a grade 3–4 southeast wind scale in a ventilated house. The sampling for each treatment was repeated three times. All samples were wrapped in tin foil, fixed by the liquid nitrogen sample-fixing method, and placed in a −70 °C refrigerator.

### RNA isolation and cDNA synthesis

Total RNA was extracted by employing the RNAprep Pure Plant Kit (Tiangen Biotech Co. Ltd., Beijing, China) on the basis of the manufacturer's explanatory memorandum. The concentration and A260–A280 ratios of total RNA were evaluated by a spectrometer

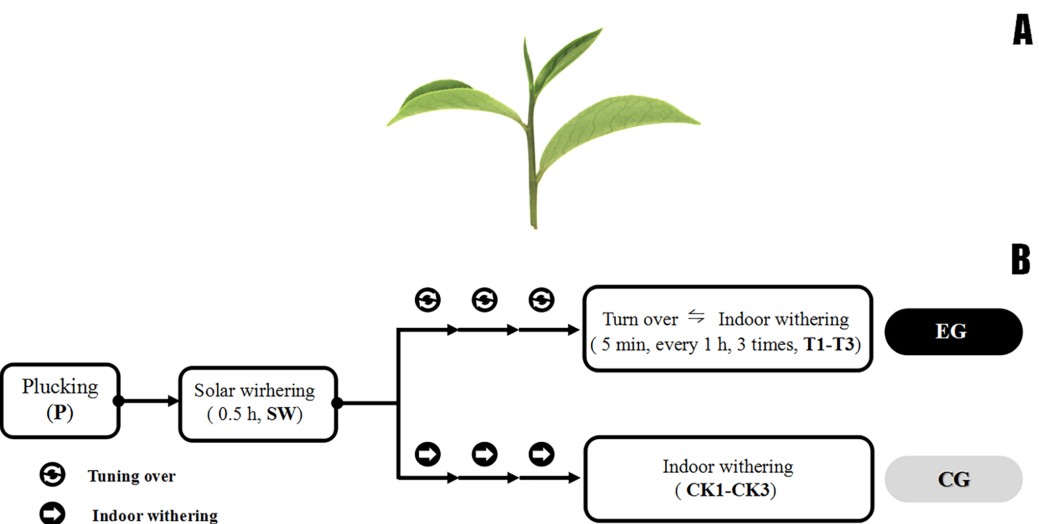

**Figure 1 Fresh tea leave and post-harvest processing steps.** (A) Photograph of the leaf tissue of the tea plant cultivar "Huangdan." The second leaf and its implicative stem from one bud and three of four leaves of tea (*Camellia sinensis* var. Hu angdan) were plucked after every post-harvest treatment. (B) The post-harvest processing steps of fresh tea leaves, solar withering made the fresh leaves (P) plucked from tea plantation become wither leaves (SW). Upstream was experimental group consisted of turn over and withering treatment (T1–T3), while the downstream was control group without turn over treatment (CK1–CK3), every sample was taken at the same time point of those in experimental group.

(Thermo Fisher, Waltham, MA, USA) to detect the purity. Total RNA mixed with RNA loading buffer was added into1.2% agarose gel on electrophoresis apparatus for 10 min at 150 V and 40 mA in fresh 1× TAE buffer. An imaging system has been implemented and tested for integrity analysis of total RNA electrophoretic films. Afterward, cDNA (20 μL) was synthesized from the normalization quality of total RNA using the PrimeScript RT Reagent Kit with a gDNA Eraser (TaKaRa Biotech Co., Ltd., Dalian, China).

## Selection of candidate reference genes and primer design

*Cs18SrRNA*, *CsGADPH*, *CsACT*, *CsEF-1α*, *CsUbi*, and *CsTUA*. *Cs26SrRNA* and *CsRuBP* were chosen from the recommendations of the report by *Gohain et al. (2011)*, and the rest of the genes—*CsCYP*, *CselF-4α*, *CsMON1*, *CsPCS1*, *CsSAND*, *CsPPA2*, *CsTBP* were picked based on the TAIR database (*Wang et al., 2017*). The RT-qPCR primers were designed with DNAMAN 7.0 (Lynnon BioSoft, San Ramon, CA, USA) (Table 1).

## Quantitative real-time PCR assay

The RT-qPCR reactions were performed using a LightCycle® 480 Real-Time PCR System (Roche, Indianapolis, IN, USA). Each amplification in a 96-well plate was performed in a 20 μL final volume containing 10 μL of 2 × SYBR Premix Ex Taq™ (TaKaRa); 0.8 μL of each specific primer pair at 200 nM; 1.0 μL of 4 × diluted cDNA template (300 ng/μL); and 7.4 μL of ddH$_2$O. The PCR reaction conditions were as follows: denaturation for 10 s at 95 °C, 40 cycles of 5 s at 95 °C, and 20 s between 55 and 60 °C using the $T_m$ function of the primers. Fluorescent detection was performed after each

**Table 1 The characteristics of primers of candidate reference genes for RT-qPCR of *Camellia sinensis*.**

| Gene symbol | Accession number or Arabidopsis homolog locus | Forward/reverse primer sequence (5′–3′) | Amplicon length (bp) | Melting temperature (°C) | Efficiency value (%) | $R^2$ |
|---|---|---|---|---|---|---|
| Cs18SrRNA | AY563528.1 | CGGCTACCACATCCAAGGAA/ GCTGGAATTACCGCGGCT | 191 | 63.2/63.5 | 95.6 | 0.994 |
| CsGADPH | KA295375.1 | TTGGCATCGTTGAGGGTCT/ CAGTGGGAACACGGAAAGC | 206 | 61.6/61.7 | 100.3 | 1.000 |
| CsACT | KA280216.1 | GCCATCTTTGATTGGAATGG/ GGTGCCACAACCTTGATCTT | 175 | 60.3/60.0 | 104.5 | 0.980 |
| CsEF-1α | KA280301.1 | TTCCAAGGATGGGCAGAC/ TGGGACGAAGGGGATTTT | 196 | 59.6/60.2 | 99.2 | 0.999 |
| CsUbi | HM003234.1 | GGAAGGACTTTGGCTGAC/ GACCCATATCCCCAGAACAC | 98 | 55.6/59.1 | 92.3 | 0.992 |
| CsTUA | DQ444294.1 | TCCAAACTAACCTTGTGCCATAC/ ACACCCTTGGGTACTACATCTCC | 220 | 60.3/60.5 | 97.5 | 0.983 |
| Cs26SrRNA | AY283368 | TCAAATTCCGAAGGTCTAAAG/ CGGAAACGGCAAAAGTG | 319 | 56.2/58.8 | 95.6 | 0.984 |
| CsRuBP | EF011075.1 | AAGCACAATTGGGAAAAGAAG/ AAAGTGAAAATGAAAAGCGACAAT | 405 | 58.4/60.4 | 103.4 | 0.999 |
| CsCYP | AT3G56070 | TTTGCGGATGAGAACTTCAA/ CCATCTCCTTCACCACACTG | 181 | 59.4/59.1 | 104.5 | 0.997 |
| CseIF-4α | AT3G13920 | TGAGAAGGTTATGCGAGCAC/ GCAACATGTCAAACACACGA | 149 | 59.0/59.1 | 109.0 | 0.989 |
| CsMON1 | AT2G28390 | ATTTCCTTCGTGGAGAATGG/ GCCCATAAACAAGCTCCAAT | 160 | 59.0/59.0 | 92.1 | 0.986 |
| CsPCS1 | AT5G44070 | AATGCCCTTGCTATTGATCC/ CTCCAGAACAGTGAGCCAAA | 151 | 59.0/59.0 | 98.9 | 0.981 |
| CsSAND | AT2G28390 | GCCTGAACCGTCTTCTGTGGAGT/ CTCAATCTCAGACACACTGGTGCTA | 184 | 66.2/63.0 | 100.6 | 0.984 |
| CsPP2A | AT3G21650 | AAGAAGAGGAACTGGCGACGGAAC/ CAAACAGGTCCAGCAAACGCAAC | 153 | 67.9/68 | 95.7 | 0.993 |
| CsTBP | AT1G55520 | GGCGGATCAAGTGTTGGAAGGGAG/ ACGCTTGGGATTGTATTCGGCATTA | 166 | 68.0/68.1 | 97.6 | 0.987 |
| CsLOX1 | EU195885.2 | AACAAGAACAACAATATATAGCTC/ AAACGGAGCCTTCAACACC | 165 | 51.8/60.1 | 101.1 | 0.994 |

extension step. The electrophoresis method was used to detected DNA bands of candidate reference genes from PCR production with 2% agarose gel. Each assay included three technical repetitions and involved a standard curve with five serial dilution point with ddH$_2$O (gradient was as follow: 1:1, 1:5, 1:25, 1:125, and 1:625), which were set to calculate PCR efficiency and obtain the suitable annealing temperature (*Vandesompele et al., 2002*). To validate the normalization effect of cDNA templates and the stability of candidate reference genes, *CsLOX1* was selected as a calibration gene. The expressions of *CsLOX1* were tested under the same RT-qPCR conditions, except with annealing temperatures of 60 °C.

| Table 2 | The concentration and A260/A280 ratios of total RNA of samples. | | | | | | | |
|---|---|---|---|---|---|---|---|---|
| Sample | *F* | SW | T1 | T2 | T3 | CK1 | CK2 | CK3 |
| Concentration (ng/μL) | 675.5 | 630.2 | 703.7 | 881.9 | 750.3 | 881.9 | 823.0 | 798.3 |
| A260/A280 | 2.09 | 2.08 | 2.05 | 2.05 | 2.07 | 2.09 | 2.03 | 2.07 |

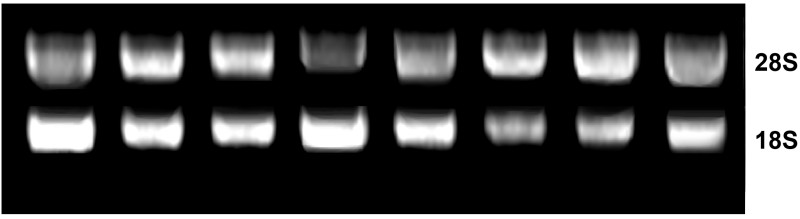

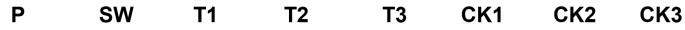

| P | SW | T1 | T2 | T3 | CK1 | CK2 | CK3 |

**Figure 2 The electrophoretogram of total RNA of Huangdan samples during the post-harvest processing steps.** The clear 18S bands and 28S bands of Huangdan samples during the manufacturing could be distinguished, which indicated the purity of total RNA of each sample was good. Samples were selected at each of step during the tea leaves post-harvest. F, fresh leave; SW, solar withered leaves; T1, T2, and T3, leaves after the each turn over treatment; CK1, CK2, and CK3, leaves after the each withering treatment.

## Data analysis

GeNorm v3.5, NormFinder v0.953, and Bestkeeper v1.0 were utilized to evaluate the expression stability of the candidate reference genes. The expression pattern of *CsLOX1* gene was calculated based on the normalization factors (NFs), a series of coefficients and produced by GeNorm. Additionally, PASW Statistics v18.0 were used to analyze the difference of gene expression levels.

## RESULTS

### Evaluation of total RNA quality

The method of centrifugal column was utilized to purify total RNAs of leaf samples during post-harvest processing steps. The result of total RNA quality detection indicated that the concentrations of all sample were greater than 500 ng/μL and the A260/A280 ratios were between 2.00 and 2.10 (Table 2). The result of agarose gel electrophoresis showed that 18S bands and 28S bands of each sample were clear, uniform, and separated distinctly (Fig. 2). On the whole, although the fresh tea leaves have been in vitro state for a while, the total RNA of all samples maintained a good purity and integrity. That meant the level of gene transcription still existed, which layed the foundation of RT-qPCR assay as follow.

### Evaluation of primer specificity and amplification efficiency

Based on the sequences of 15 candidate reference genes cloned in a previous study (*Sun et al., 2010*; *Gohain et al., 2011*; *Wang et al., 2017*; *Wu et al., 2016a*; *Pothinuch &*

*Tongchitpakdee, 2011*), gene-specific primer pairs were design. The specific circumstances, including the accession number, primer sequence, amplicon length, melting temperature, amplification efficiency (between 90.0% and 110.0%), and $R^2$ (ranged from 0.980 to 1.000) were summarized in Table 1. Regarding these genes, the cDNA of fresh tea leaves that was fixed in the tea field by liquid nitrogen as a template were utilized and the single PCR production of anticipated size was amplified with 1.0% agarose gel electrophoresis. An RT-qPCR assay was carried out by virtue of the high specificity of the amplification reaction (Fig. 3A). In addition, the melting curve analysis illustrated that there was a distinct peak for each set of primers (Fig. S1A). The standard curve analysis showed that the proper annealing temperatures and efficiency value of each set of primers was between 90% and 110% (Fig. 3B).

## Expression profiles of candidate reference genes

The $C_t$ values reflected the fluorescent signal strength which reached above the baseline threshold. A preliminary overview of the variation among 15 candidate genes was discerned from the analysis of the original expression levels in all post-harvest tea leaves (Fig. 4). There were obvious differences in the transcription abundance of the 15 genes. The $C_t$ values of these candidate genes ranged from 5.00 to 28.62 using the turn over treatment and 5.67 to 28.62 under the withering treatment. A large portion of these $C_t$ values under post-harvest treatments were between 18.96–25.08. The genes encoding *Cs18SrRNA* and *Cs26SrRNA* showed higher levels of expression compared to the protein coding genes under the two post-harvest conditions. Among all protein coding genes, the average $C_t$ values of *CsTUA* were 25.38 and 25.56, respectively, which represented the lowest transcription abundance. The average $C_t$ values of *CsRuBP* were 21.05 and 19.61, respectively, indicating superior transcription abundance of protein coding genes under the two treatments, but the overall stability of the turn over condition was greater than that of withering. Therefore, it is clear that it is crucial to explore suitable reference gene(s) to normalize the target gene expression of tea leaves under different post-harvest conditions.

## Expression stability of candidate reference genes

Regarding the turn over treatment group, *CsTBP* was the most stable gene in the GeNorm and Normfinder algorithms, whereas it was the fourth most stable under the Bestkeeper algorithm. *CsACT* and *CsEF-1α* remained relatively constant expression across all three algorithms, and *CsTBP*, *CsACT*, *CsEF-1α*, and *CselF-4α* were all ranked among the six most stable genes. *Cs18SrRNA, CsTUA, Cs26SrRNA*, and *CsRuBP* exhibited unstable expression in all algorithms (Table 3). The average stability of the reference genes varied with the sequential addition of each reference gene to the equation (when calculating the NF).

Certainly, the stability of candidate reference genes played into the validation. However, the number of reference genes cannot be ignored. The pairwise variation ($V_n/V_{n+1}$, ($n \geq 2$)) corresponded to the reference gene numbers used to determine the NF. The $V_{2/3}$ value was shown to be 0.099, which is below the recommended cut-off value of 0.15 (*Wu et al., 2016a*) (Fig. 5A), suggesting it is unnecessary to select more than two genes to

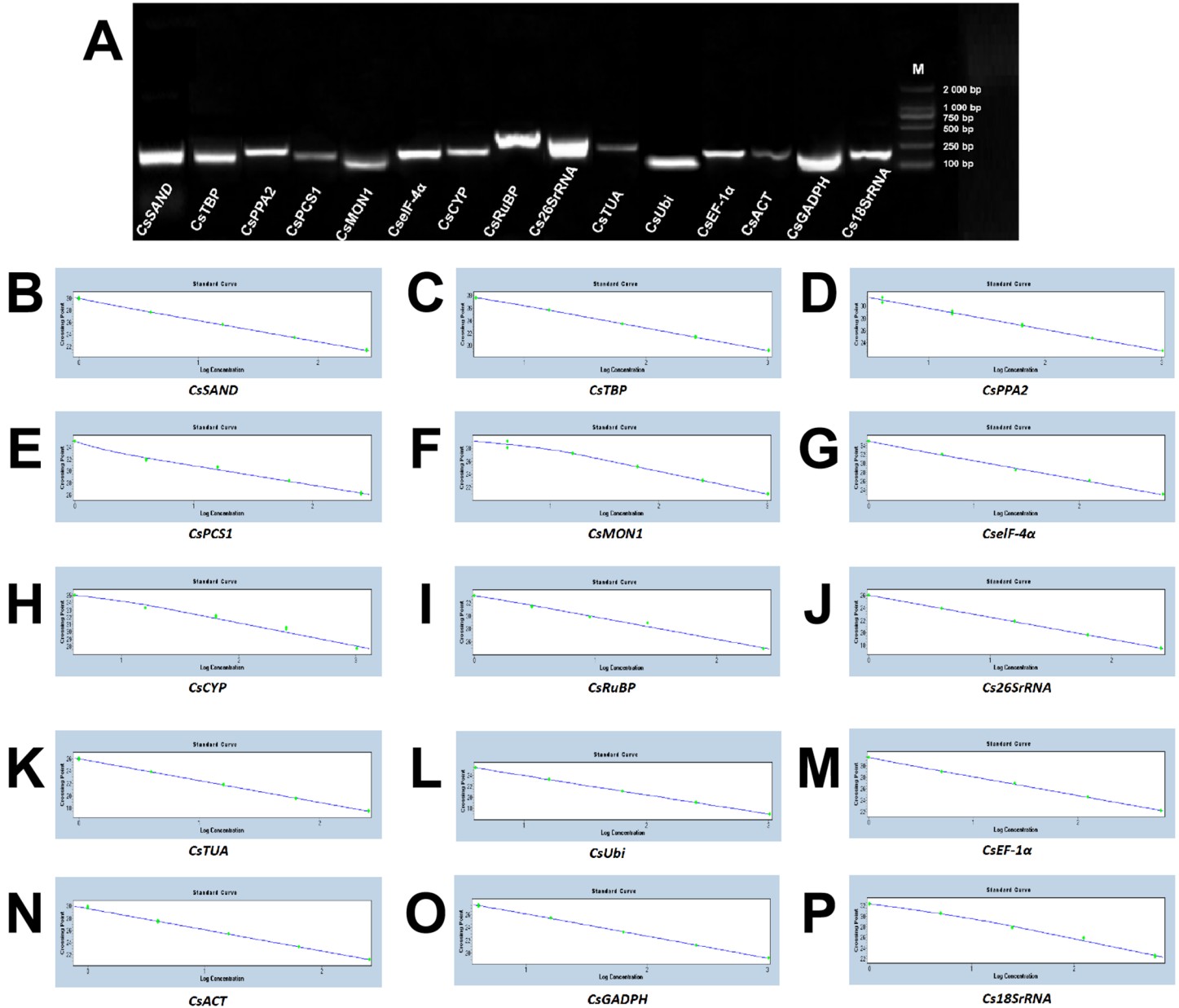

**Figure 3 Verification of primer pairs for the size of the RT-qPCR amplicon and their standard curves.** (A) Confirmation of primer specificity and amplicon size. Amplification results from 15 candidate genes using a *Camellia sinensis* cDNA template. M: DL2 000 DNA Maker. A mixture of five times was used as a gradient, which were set as templates to obtain a standard curve for each set of primers of 15 candidate genes to determine the appropriate amplification concentration, temperature and efficiency value. They were (B) CsSAND; (C) CsTBP; (D) CsPPA2; (E) CsPCS1; (F) CsMON; (G) CseIF-4α; (H) CsCYP; (I) CsRuBP; (J) Cs26SrRNA; (K) CsTUA; (L) CsUbi; (M) CsEF-1α; (N) CsACT; (O) CsGADPH; (P) Cs18SrRNA, respectively.

calculate NFs. The most stable combination was *CsPPA2* and *CsTBP* (Fig. 6A), so it is clear that these two genes could be considered to be a suitable combination.

Similar to the above, the result from the three algorithms for the withering group differed from each other (Table 4). *CsTBP*, *CsPCS1*, and *CsPPA2* were shown to be the most stable genes. Of the top four most frequent genes, only one, *CsTBP*, it was included,

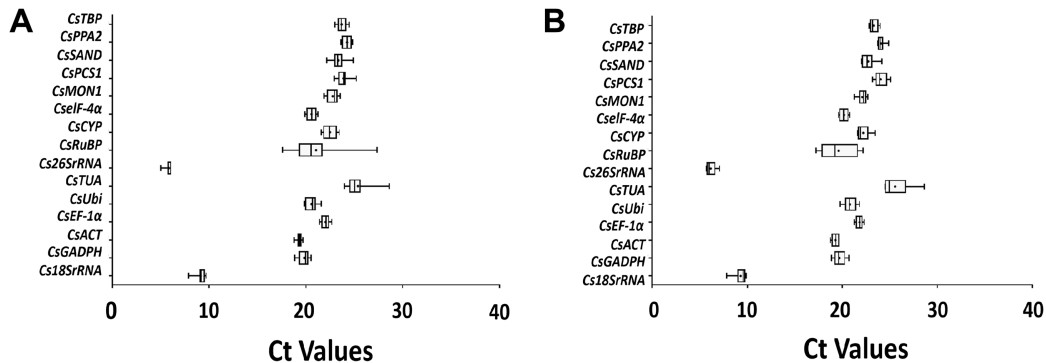

**Figure 4 RT-qPCT $C_t$ values of 15 candidate reference genes in *Camellia sinensis* leaves under turn over (A) and withering (B) treatments.** Expression data is displayed as $C_t$ values for each reference gene for all *Camellia sinensis* samples. The lines across the boxes represent the mean $C_t$ values. The boxes indicate the 25th and 75th percentiles, while the whiskers correspond to the maximum and minimum values.                                                

**Table 3 The ranking of candidate reference gene stability by three software programs during turn over treatment.**

| Rank | GeNorm | | Normfinder | | BestKeeper | | |
|---|---|---|---|---|---|---|---|
| | Gene | Stability | Gene | Stability | Gene | SD | CV |
| 1 | CsTBP | 0.051 | CsTBP | 0.005 | CsACT | 0.186 | 0.965 |
| 2 | CsACT | 0.053 | CsACT | 0.009 | CsEF-1α | 0.302 | 1.370 |
| 3 | CsEF-1α | 0.054 | CsEF-1α | 0.009 | CsPPA2 | 0.340 | 1.402 |
| 4 | CsPPA2 | 0.055 | CsGADPH | 0.015 | CsTBP | 0.363 | 1.530 |
| 5 | CselF-4α | 0.056 | CsCYP | 0.015 | CsPCS1 | 0.449 | 1.880 |
| 6 | CsMON1 | 0.056 | CselF-4α | 0.016 | CsGADPH | 0.416 | 2.095 |
| 7 | CsGADPH | 0.057 | CsPPA2 | 0.016 | CsMON1 | 0.482 | 2.116 |
| 8 | CsCYP | 0.057 | CsMON1 | 0.018 | CsUbi | 0.441 | 2.146 |
| 9 | CsPCS1 | 0.061 | CsPCS1 | 0.019 | CsSAND | 0.513 | 2.197 |
| 10 | CsUbi | 0.066 | CsUbi | 0.029 | CselF-4α | 0.494 | 2.400 |
| 11 | CsSAND | 0.075 | CsSAND | 0.037 | CsCYP | 0.570 | 2.534 |
| 12 | Cs18SrRNA | 0.096 | Cs18SrRNA | 0.055 | Cs18SrRNA | 0.096 | 3.546 |
| 13 | CsTUA | 0.099 | CsTUA | 0.060 | CsTUA | 0.912 | 3.591 |
| 14 | Cs26SrRNA | 0.107 | Cs26SrRNA | 0.063 | Cs26SrRNA | 0.249 | 4.277 |
| 15 | CsRuBP | 0.175 | CsRuBP | 0.117 | CsRuBP | 1.813 | 8.610 |

**Note:**
SD, standard deviation; CV, coefficient variation.

and *CsTBP*, *CsPCS1*, *CsPPA2*, *CselF-4α*, *CsACT*, and *CsEF-1α* coexisted in the top eight most frequent genes in the three algorithms. *CsRuBP*, *Cs18SrRNA*, *Cs26SrRNA*, and *CsTUA* also performed inconsistently, similar to the turn over group. The pair variation of $V_{2/3}$ was 0.073 (Fig. 5B), which meant there was no need to select a third gene to normalize the data. The most stable combination from the GeNorm algorithm was *CselF-4α* and *CsTBP* (Fig. 6B), which was utilized to obtain reliable NFs for the withering treatment.

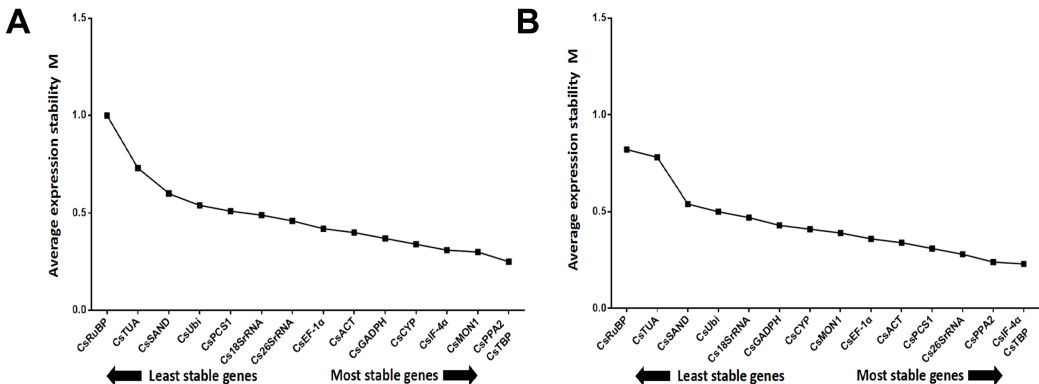

**Figure 5 Determination of the optimal number of 15 candidate genes.** Pair-wise variation (V) calculated by GeNorm to determine the minimum number of reference genes for accurate normalization in two post-harvest treatments. Arrow indicates the optimal number of genes for normalization in each sample set. (A) Turn over treatment and (B) withering treatment.

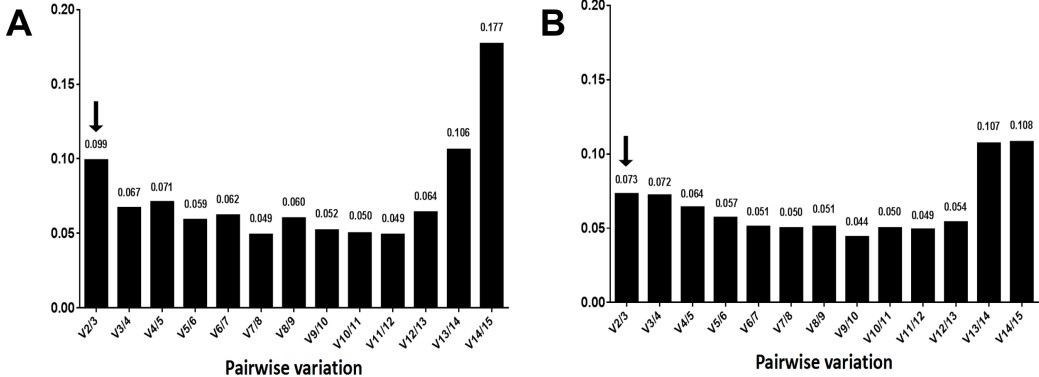

**Figure 6 Average expression stability value (M) of the 15 candidate reference genes.** Average expression stability values (M) of the reference genes were detected during stepwise exclusion of the least stable reference genes. A lower *M* value indicates more stable expression, as analyzed by the GeNorm software in the *Camellia sinensis* sample sets under two experimental conditions in post-harvest leaves of tea plants. (A) Turn over treatment and (B) withering treatment.

## Validation of selected reference genes

The *CsLOX1* gene, whose protein confers a dual positional specificity since it released C-9 and C-13 oxidized products in equal proportions, responded to the damage of external machinery to aroma formation during post-harvest of tea leaves. Based on the above consequences, the optimal NF from the most stable combination was used to normalize the expression level of *CsLOX1*. Regarding the turn over group (Fig. 7A), the expression of *CsLOX1* exhibited a general upward trend. Stages F–T1 showed a rising trend, but the expression decreased slightly at T2 due to still-standing (an essential craft during the oolong tea manufacturing process). The expression of *CsLOX1* in the second stage, from T2 to T3, increased greatly, and this was the critical period for aromatic compound formation. In contrast, there was no obvious change in the withering group before CK2, but the expression level of CK3 was very close to T3. Unlike for turn over, the middle stage,

**Table 4 The ranking of candidate reference gene stability by three software programs during withering treatment.**

| Rank | GeNorm | | Normfinder | | BestKeeper | | |
|------|--------|-----------|------------|-----------|------------|-------|-------|
| | Gene | Stability | Gene | Stability | Gene | SD | CV |
| 1 | *CsTBP* | 0.047 | *CsPCS1* | 0.006 | *CsPPA2* | 0.274 | 1.137 |
| 2 | *CsPCS1* | 0.048 | *CsTBP* | 0.007 | *CsEF-1α* | 0.278 | 1.272 |
| 3 | *CsPPA2* | 0.050 | *CsACT* | 0.013 | *CsACT* | 0.283 | 1.469 |
| 4 | *CseIF-4α* | 0.050 | *CseIF-4α* | 0.014 | *CsTBP* | 0.348 | 1.492 |
| 5 | *CsACT* | 0.051 | *CsPPA2* | 0.016 | *CseIF-4α* | 0.349 | 1.578 |
| 6 | *CsEF-1α* | 0.053 | *CsCYP* | 0.017 | *CsMON1* | 0.353 | 1.747 |
| 7 | *CsCYP* | 0.055 | *CsEF-1α* | 0.019 | *CsPCS1* | 0.475 | 1.975 |
| 8 | *CsMON1* | 0.056 | *CsGADPH* | 0.020 | *CsSAND* | 0.523 | 2.299 |
| 9 | *CsGADPH* | 0.057 | *CsMON1* | 0.024 | *CsCYP* | 0.550 | 2.476 |
| 10 | *CsUbi* | 0.063 | *CsUbi* | 0.029 | *CsGADPH* | 0.490 | 2.488 |
| 11 | *CsSAND* | 0.069 | *CsSAND* | 0.037 | *CsUbi* | 0.549 | 2.637 |
| 12 | *CsTUA* | 0.097 | *CsTUA* | 0.059 | *CsTUA* | 1.163 | 4.548 |
| 13 | *Cs26SrRNA* | 0.101 | *Cs26SrRNA* | 0.060 | *Cs18SrRNA* | 0.463 | 4.968 |
| 14 | *Cs18SrRNA* | 0.103 | *Cs18SrRNA* | 0.062 | *Cs26SrRNA* | 0.394 | 6.400 |
| 15 | *CsRuBP* | 0.123 | *CsRuBP* | 0.080 | *CsRuBP* | 1.522 | 7.758 |

**Note:**
SD, standard deviation; CV, coefficient variation.

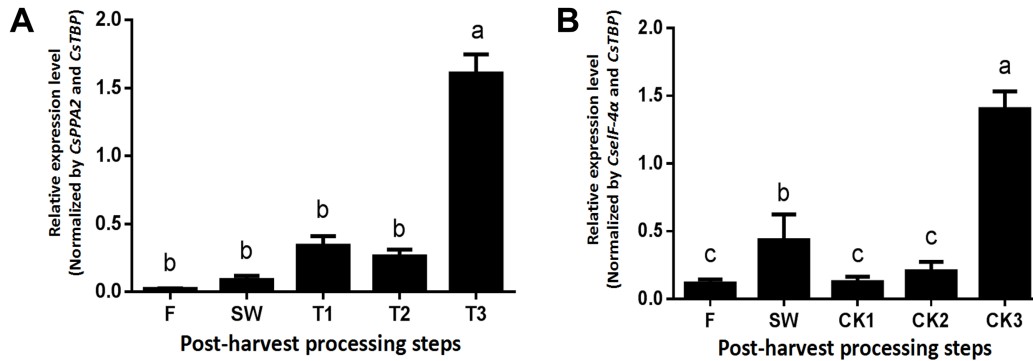

**Figure 7 Relative quantification of *CsLOX1* expression utilizing different reference genes beneath the turn over and withering treatments.** Samples were selected at each of step during the tea leaves post-harvest. F, fresh leave; SW, solar withered leaves; T1, T2, and T3, leaves after the each turn over treatment, CK1, CK2, and CK3, leaves after the each withering treatment. (A) Turn over treatment normalized by the combination of *CsPPA2* and *CsTBP*. (B) Withering treatment normalized by the combination of *CseIF-4α* and *CsTBP*. Different letters (a–c) represented significant differences at $P < 0.05$ by LSD test.

from CK1–CK2, had a steady state while the variation trend beginning at F–SW and ending at CK2–CK3 was consistent with the turn over group (Fig. 7B).

# DISCUSSION

An increasing number of gene expression pattern analyses, including for the tea tree plant, have given a new level of insight into biological phenomena. Thus, the selection of a reference gene that directly affects the final outcome is important.

 

The current experiment obtained 15 reference genes, primarily from two sources: steadily expressed genes from previous reports and the Arabidopsis information resource. We obtained the proper melting temperatures and efficiency values from standard curves, helping us to make a lot of sense to the data we got. There are some reports about essential genes expression variation in a secondary metabolic pathway during the tea leaves post-harvest (*Cho et al., 2007*; *Gui et al., 2015*; *Fu et al., 2015*).

According to the $C_t$ values of both the turn over and withering group, it was shown that most of the candidate genes, except the Cs18SrRNA and Cs26SrRNA genes, were stable at 20–25 cycles. Based on the standard deviation of the $C_t$ value, the *CsACT* gene had the lowest degree of dispersion, regardless of whether it was turned over or withered, whereas the *RuBP* gene dispersed the most under both treatments. The results of the three algorithms were close to each other after normalization. Regarding the turn over group, the *CsTBP*, *CsACT*, *CsPPA2*, and *CsEF-1α* genes all showed stable expression. *CsTUA*, *Cs18SrRNA*, *Cs26SrRNA*, and *CsRuBP* were not stable in different algorithms. The *CsMON1*, *CsUbi* and, *CsSAND* genes showed an average level of stability. However, the evaluation of different algorithms for the stable genes using the turn over treatment varied compared to the withering group. Considering the withering group, the *CsTBP*, *CsACT*, and *CselF-4α* genes were still the most stable. *CsTUA*, *Cs18SrRNA*, and *Cs26SrRNA* were expressed in an unstable manner, which was consistent with the turn over treatment. Additionally, *CsMON1*, *CsUbi*, and *CsGADPH* had average stability.

The pioneers, *Sun et al. (2010)* investigated the most stable reference genes in different organs and tissues and drew the conclusion that β-ACT performs well in organs and *CsGADPH* was suitable for mature leaves and callus. Lately, a number of selected reference genes have been observed for the tea plant. Similar to the current research, by utilizing five developmental stages of tea leaves, *Wu et al. (2016a)* found that *CsTBP* and *CsTIP4* is the best combination, and *CsTBP* also plays a very stable role under different hormonal stimuli treatments. *Wang et al. (2017)* picked 12 candidate genes to determine the most suitable reference gene under different mental stresses, and found that *CsPP2AA3* and *Cs18Sr RNA* were the most stably expressed genes and *CsGAPDH* and *CsTBP* were the least stable. Moreover, *Hao et al. (2014)* researched the most stably expressed reference genes of the tea tree in different time periods, including its harvest by auxin and lanolin among 11 candidate reference genes in 94 experimental samples, they found that the top five appropriate reference genes were *CsPTB1*, *CsEF1*, *CsSAND1*, *CsCLATHRIN1*, and *CsUBC1* under experimental conditions. *Gohain et al. (2011)* identified the most suitable reference gene, *CsRuBP*, that ran, counter to the current result, under different experimental conditions, mainly biotic and abiotic stresses.

The stable reference genes screened from turn over group were equivalent to that of hormone treatment (*Wu et al., 2016a*) and metal stress (*Wang et al., 2017*). However, the most suitable genes selected by *Gohain et al. (2011)*, *Sun et al. (2010)*, and *Chen et al. (2017)* are quite different from what we analyzed. We considered that this was mainly due to the differences of physiological characteristics in the pre- and post- harvest, and the inconsistency of tea varieties might also work. Previously, studies have used different reference genes under different biotic and abiotic stress treatments. *Liu & Han (2010)*

used *CsGADPH* as a reference gene to research the expression patterns of *CsLOX1* in different tissue parts and during the open and senescence stages of tea plants, for instance. *Fu et al. (2015)* utilized *CsEF-1α* as a reference gene to study the changes in genes related to aroma formation in different metabolic pathways. *Cho et al. (2007)* used *Cs26SrRNA* as a reference gene to construct an aroma-related gene expression profile during the manufacturing process of Oriental Beauty. However, the results of the turn over group and withering group in the current study show that the most suitable reference is not the same under different treatments which imitated the crafts and tea varieties. Thus, when selecting suitable reference genes to transform test materials, space and time have a large impact. Referring to the plant in vivo under different stress inductions is not enough, only making a correction to the reference genes of the tea leaf in vitro can pinpoint and then construct a tea leaf expression pattern of a reliable target gene precisely.

*CsLOX1*, a key gene in the fatty acid metabolic pathway of *C. sinensis*, has a double cleavage site (9/13-) (*Liu & Han, 2010*). An existing study found that the activity of LOX increases during the turning process (*Hu et al., 2018*). *Zeng et al. (2018)* found that *CsLOX1* is significantly affected by multiple environmental stresses and involves the biosynthesis of jasmine lactone during the tea manufacturing process. Hence, it could be used as an ideal gene for verifying the reference genes under post-harvest treatment. *CsLOX1* responded to external mechanical damage and was significantly increased in the turn over group due to the acceleration of water waste, thereby contributing to the formation of rich aromatic substances, which was similar to the result of *Gui et al. (2015)* and *Zeng et al. (2018)*, while in the withering group, *CsLOX1* was at a low level without mechanical stimulation. However, the loss of water reached a certain level after overnight storage (T3), inducing the up-regulation of *CsLOX1* expression, which was close to that of *Wang et al. (2019)*. Therefore, we can make it clear that the most suitable combination of reference genes we selected were feasible from the *CsLOX1* expression pattern.

## CONCLUSIONS

This research, for the first time, screened reference genes during the process of post-harvest treatment of tea plant leaves (oolong tea varieties), filling the gaps of suitable reference genes during manufacturing process of oolong tea. Through a systematic analysis and research, we found that when selecting a single reference gene for normalization, whether for the turn over treatment or the withering treatment, it was advisable to choose *CsTBP*. When multiple reference genes were used for normalization, the *CsPPA2* and *CsTBP* genes were suitable for turn over treatment, and the combination of *CseIF-4α* and *CsTBP* should be selected during withering treatment. We hold the opinion that the result normalized by multiple reference genes was more accurate than that of single gene. On the other hand, the suitable reference genes we selected might also be used in some other horticulture plant during the post-harvest treatments. We expect that our conclusions could be used for related researches to obtain more accurate and reliable data of gene relative expression level.

### Funding

This work was supported by the Earmarked Fund for China Agriculture Research System (CARS-19) and the Major Science and Technology Project in Fujian Province (2015NZ0002-1). The funders had no role in study design, data collection and analysis, decision to publish, or preparation of the manuscript.

### Grant Disclosures

The following grant information was disclosed by the authors:
Earmarked Fund for China Agriculture Research System: CARS-19.
Major Science and Technology Project in Fujian Province: 2015NZ0002-1.

### Competing Interests

The authors declare that there is no conflict of interests regarding the publication of this article.

### Author Contributions

- Zi-wei Zhou performed the experiments, analyzed the data, prepared figures and/or tables, authored or reviewed drafts of the paper, approved the final draft.
- Hui-li Deng performed the experiments, analyzed the data, prepared figures and/or tables.
- Qing-yang Wu performed the experiments, analyzed the data, prepared figures and/or tables.
- Bin-bin Liu performed the experiments, analyzed the data, prepared figures and/or tables.
- Chuan Yue contributed reagents/materials/analysis tools, authored or reviewed drafts of the paper.
- Ting-ting Deng contributed reagents/materials/analysis tools.
- Zhong-xiong Lai conceived and designed the experiments, authored or reviewed drafts of the paper, approved the final draft.
- Yun Sun conceived and designed the experiments, authored or reviewed drafts of the paper, approved the final draft.

### Data Availability

The raw data is provided in Fig. S1 and Table S1.

### Supplemental Information

Supplemental information for this article can be found online at http://dx.doi.org/10.7717/peerj.6385#supplemental-information.

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
