# Peer review of "Validation of reference genes for gene expression studies in post-harvest leaves of tea plant (Camellia sinensis)"

_PeerJ, doi:10.7717/peerj.6385_

## Round 0.1 · original submission · Major Revisions

The reviewers had some remarks, and one attached a manuscript with suggested correction. Please consider their comments. I believe the paper needs moderate revision, rather than major revision. I look forward to you submitting an updated manuscript.

·

Basic reporting

1- The English writing needs to revise. Also, some correction should be done for some incorrect words inside the text.
2- In Abstract section - lines 54 and 55: while CsTBP, CsPCS1, CsPPA2, CselF-4α, and
CsACT were the four best reference genes in the witheringgroup.
They are 5 and not 4. Please correct.
3- 18S rRNA and something like this should not be written in italic. In Discussion section, they are italic. Please correct them.
4- In Material and Methods - lines 141 and 142 has been included the results. Please put the numbers regarding to results inside the Results section.
5- Please explain that why did use agarose gel electrophoresis in 1.2%?
6-The current more that 50 mA could make heating inside the gel that could degrade RNA. Please rewrite this section (line 144).
7-In title of "Quantitative Real Time assay", line 161, agarose gele electrophoresis is not considered as quantitative analysis. Please insert the results of standard curves that mentioned inside the text.
8-Inside the Table 1, three genes has been shown but in its description has been written 4 genes. Which of them is correct?
9-Inside the Table regarding to primer pairs, The amplicons more than 180 to 200 bp is not suitable for Real time analysis. They are acceptable for publishing. The primer designing should be done again and this section should be repeated.

Experimental design

The primer pairs with amplification of more than 200 bp should be repeated in order to re-designing the primers and then do qRT-PCR.
The standard curve should be added and discussed more.

Validity of the findings

The authors should explain why the previous reference genes are not suitable and why they chose their interested genes?

·

Basic reporting

Clear and unambiguous, professional English used throughout.

Experimental design

Research question well defined, relevant & meaningful. It is stated how research fills an identified knowledge gap.

Validity of the findings

Data is robust, statistically sound, & controlled.

Additional comments

The manuscript "Validation of reference genes for gene expression studies in post-harvest leaves of tea plant" by ZhOU et al. sumamrizes a good and solid research endeavor that tested the reference genes suitable for the quantitative real-time PCR analysis of gene expression in post-harvest leaves of tea plant.

In a word, the manuscript was well-prepared, especially it is written in professional English, which makes understanding easier for readers. The manuscript is technically sound, and the data supports the conclusions. Besides, the statistical analysis been performed appropriately and rigorously.

0verall, the present version meet the criteria of Peer J, however the revision include but not limit to the following should be needed before it can be reconsidered for publication (For detailed information, please see the attached review commented file). Therefore, I recommend the publication with minor revision.

1. When you describe the plant material, what do you mean by "second"? From the top or from the bottom? And you also need to confirm that it is a scientific and reasonable description on the leaf position.
2. In the parts of "Materials & Methods", you should not describe know the results, such as “The concentration (≥ 500142 ng/μL) and A260–A280 ratios (1.90-2.10) of total RNA were evaluated by a spectrometer 143 (Thermo USA) to detect the purity.” In lines 141-143.
3. How many candidate reference genes have you selected? 15 or 16 (8=8)? If CsTIP41 is redundant? If so, you should deleted all its related information, including in the Table.
4. Regarding the “Validation of Selected Reference Genes”, you are suggested to highlight how the results of CsLOX1 gene expression normalized by the selected reference genes accord with what it is assumed to be, which is the most direct evidence on the feasibility of the selected reference genes.
5. Several references are missing, and you need to add the corresponding references accordingly.
6. Please keep in mind that, in the Parts of Discussion, de-emphasize the findings that have been previously reported by others and the differences between this research and previous researches. Especially, perform a critical discussion of the main results and novel contributions of this article, and how this adds to the current knowledge status in the field.

---

## Round 0.2 · accepted · Accept

The reviewer who had critical remarks at first version recommended accept the manuscript in current form. The other reviewer had no more critical comments after the revision, as they wrote me.

Happy New Year!

# ·

Basic reporting

no comment

Experimental design

no comment

Validity of the findings

no comment

Additional comments

All the changes are acceptable.

Reviewer 3 ·

Basic reporting

Good article

Experimental design

Correct

Validity of the findings

Valid

Additional comments

No comments